

# Performance enhancement in hydroponic and soil compound prediction by deep learning techniques

Mustufa Haider Abidi[1], Sanjay Chintakindi[2], Ateekh Ur Rehman[2] and Muneer Khan Mohammed[1]

[1] Advanced Manufacturing Institute, King Saud University, Riyadh, Saudi Arabia
[2] Department of Industrial Engineering, King Saud University, Riyadh, Saudi Arabia

Corresponding author
Mustufa Haider Abidi,
mabidi@ksu.edu.sa

## ABSTRACT

The soil quality plays a crucial role in providing essential nutrients for crop growth and ensuring a bountiful yield. Identifying the soil composition, which includes sand, silt particles, and the mixture of clay in specific proportions, is vital for making informed decisions about crop selection and managing weed growth. Furthermore, soil pollution from emerging contaminants presents a substantial risk to water resource management and food production. Developing numerical models to comprehensively describe the transport and reactions of chemicals within both the plants and soil is of utmost importance in crafting effective mitigation strategies. To address the limitations of traditional models, this paper devises an innovative approach that leverages deep learning to predict hydroponic and soil compound dynamics during plant growth. This method not only enhances the understanding of how plants interact with their environment but also aids in making more informed decisions about agriculture, ultimately contributing to more sustainable and efficient crop production. The data needed to perform the developed hydroponic and soil compound prediction model is acquired from online resources. After that, this data is forwarded to the feature extraction phase. The weighted features, deep belief network (DBN) features, and the original features are achieved in the feature extraction stage. To get the weighted features, the weights are optimally obtained using the Iteration-assisted Enhanced Mother Optimization Algorithm (IEMOA). Subsequently, these extracted features are fed into the Multi-Scale feature fusion-based Convolution Autoencoder with a Gated Recurrent Unit (MS-CAGRU) network for hydroponic and soil compound prediction. Thus, the hydroponic and soil compound prediction data is attained in the end. Finally, the performance evaluation of the suggested work is conducted and contrasted with numerous conventional models to showcase the system's efficacy.

## INTRODUCTION

### Background of the study

Soil plays a vital role in sustaining human survival and facilitating overall ecosystem health. However, the relentless pursuit of economic development and advancements in agricultural production has exacerbated the issue of soil pollution. Among the various sources of soil contamination, heavy metal pollution stands out as a particularly challenging problem due to its resistance to microbial degradation (*Kim, Grunwald & Rivero, 2014*). This contamination not only hampers crop growth and leads to reduced yields, but also poses a direct threat to human life and well-being as heavy metals can find their way into the human body through food consumption and other pathways, thereby introducing both carcinogenic and noncarcinogenic risks to the health of human, primarily when vegetables are grown in contaminated soil (*Tan et al., 2021*). It is imperative to conduct comprehensive research on heavy metal pollution in soil (*Vohland et al., 2016*). In recent years, increased attention has been devoted to the issue of heavy metal pollution in soil, leading to more profound and comprehensive studies in this field. Worldwide, the humid tropics' soils have long been favored for agriculture. However, the soils in southeastern Nigeria present unique challenges while holding potential for crop production. These soils have been severely weathered and leached due to high rainfall and high-temperature conditions, resulting in distinctive characteristics (*Wang, Tao & Zou, 2020*). Like all soils, their properties, such as texture, pH, Cation Exchange Capacity (CEC), exchangeable cations, and clay content, are shaped by various environmental factors, including topography and other soil-forming influences (*Loew & Mauser, 2008*). Additionally, these alluvial deposits' nutrient composition, mineralogical features, and soil texture are primarily characterized by quartz oxides, which are typically lacking in essential plant nutrients (*Wang et al., 2022*). Consequently, achieving high crop yields on these soils can be challenging unless appropriate nutrient amendments are applied (*Shi et al., 2022*).

Soil texture plays a crucial role in influencing the movement and storage of water and air within the soil, as well as impacting root growth, the availability of plant nutrients, and the activities of microorganisms. Collectively, these factors affect quality, overall health, and soil fertility (*Lee, 2021*). Therefore, determining and classifying soil texture is vital for a range of decision-making processes, monitoring tools, and agricultural applications (*Southey et al., 2015*). In pursuit of these goals, Visible and Near-Infrared (Vis-NIR) spectra spanning from 400 to 2,500 nm and Mid-Infrared (MIR) spectra from 2,500 to 25,000 nm have become widely adopted for assessing soil properties. These spectral techniques offer the advantage of being rapid, convenient, and cost-effective for obtaining valuable soil information (*Wang et al., 2023*). Soil is the primary reservoir for a wide range of organic contaminants (*Burton et al., 2015*). The migration of these organic contaminants from the soil into plants, especially agricultural crops, poses a significant risk to the sustainability of agriculture and the potential for negative impacts on human health through dietary exposure. This has highlighted the necessity for the development of dependable predictive tools capable of assessing the transfer of contaminants from soil to trees. These tools would

be based on these chemicals' molecular characteristics or signatures (*Jadidoleslam et al., 2022*).

## Motivation of the study

Machine learning and deep learning approaches have found extensive applications in diverse fields, such as natural language processing, healthcare, manufacturing, chemistry, and image recognition, including tasks like reaction prediction and molecular property estimation (*Cao & Zhang, 2020*; *Abidi et al., 2023a*; *Abidi, Alkhalefah & Umer, 2022*; *Abidi et al., 2023b*; *Abidi, Alkhalefah & Aboudaif, 2024*). In recent times, as part of the big data-driven toolkit for assessment and decision-making, machine learning models have demonstrated their effectiveness in predicting various characterization parameters. While numerous machine learning algorithms have been created for soil property prediction, the development of site-specific techniques is essential for improving the quality of thematic soil maps. One method to mitigate soil heterogeneity and lead to varied crop yields is using digital soil mapping (DSM). However, DSM is often hindered by within-site variability. Addressing these challenges has led to the development of site-specific cropping systems, often called precision agriculture (*Goldman et al., 2020*). Precision agriculture techniques allow for the precise delineation of management strategies for specific areas within a field. This approach has evolved to incorporate the spatial variation of nutrients and soil properties within a field, leveraging geospatial technologies and incorporating data from remote sensing, soil properties, micro-climatic data, geological information, and digital elevation models (DEM) (*Abdulraheem et al., 2023*). Precision agriculture empowers farm managers to effectively address within-field variability and maximize the cost-effectiveness of their proposed crop enterprises. In existing research works, the diverse techniques were adapted for the prediction of hydroponic and soil compounds in the plant growth. Here, the technique provides advantageous performance still it lacks from several challenges needs to be resolved. Further, the existing techniques do not efficient in the larger datasets yet, it often does not handle the complex datas effectively. In the aforementioned challenges, the novel technique is implemented in the hydroponic and soil compound prediction model. However, the implementation is done and also the datas are collected from the Kaggle dataset to provide better performance in this research work. Moreover, the result analysis shows better performance in terms of accuracy.

Some of the objectives of the suggested deep learning-based hydroponic and soil compound prediction technique are given here:

- To design a deep learning-based model for hydroponic and soil compound prediction approach that helps to make more informed choices about crop production, such as adjusting nutrient levels, irrigation schedules, and environmental conditions to optimize plant growth.
- To obtain three diverse features like input feature, weighted feature, and deep feature for prediction. These diverse feature sets can enhance the performance and interpretability of these models.

- To propose the optimization approach named IEMOA (Iteration-assisted Enhanced Mother Optimization Algorithm), which is favorable for optimizing the weight and maximizing the chi-squared statistic and relief score.
- To design the MS-CAGRU (Multi-Scale feature fusion-based Convolution Autoencoder with Gated Recurrent Unit) network for compound prediction, which is the combination of multiple powerful techniques in deep learning like convolutional autoencoders, GRU, and multi-scale feature, this model empowers the crop growth in both hydroponic and soil-based systems.
- Assessing the developed model's performance involves evaluating its predictive accuracy, generalization capabilities, and efficiency. The model's performance is likely compared to that of traditional models to demonstrate its superiority.

The main stages of the recommended hydroponic and soil compound prediction technique model utilizing deep learning are outlined in the following sections. The conventional works regarding hydroponic and soil compound prediction are explained in sub-division 2. 'An Automated Model of Hydroponic and Soil Compound Prediction during Plant Growth using Adaptive Technique' presents the definition of the automated model of hydroponic and soil compound prediction during plant growth using adaptive technique. Extracting the features and weighted features using deep belief network (DBN) and IEMOA with the objective derivation using the suggested techniques are provided in 'Extracting the Features and Weighted Features using DBN and IEMOA with the Objective Derivation'. 'Multiscale Feature Fusion-based Convolutional Autoencoder with GRU for Prediction' provides an explanation of the multistate feature fusion-based convolution autoencoder with GRU used for prediction. 'Simulation Findings and Discussions' presents numeric outcomes and offers a summary of the deep learning-driven method for forecasting hydroponic and soil compound dynamics. Finally, conclusions are presented in 'Conclusions'.

## LITERATURE REVIEW

This section explains the existing literature in the field of application of deep learning for farming, specifically for soil analysis.

### Related works

*Gao et al. (2021)* have suggested end-to-end machine learning methods to unravel the intricate relationship between complex molecular structures and risk characterization factors (RCF). These models were trained on a comprehensive RCF dataset comprising 341 data points encompassing 72 different chemicals. Our study showcased the effectiveness of the Gradient Boosting Regression Tree (GBRT) method, which was based on Extended Connectivity Fingerprints (ECFP), in forecasting RCF values. Furthermore, our findings shed light on the presence of nonlinear relationships among the properties of soils, plants, and chemicals. It aided in providing a more comprehensive understanding of the risks posed by chemicals to both human health and ecosystems.

*Emadi et al. (2020)* have employed various kind of approaches, including artificial neural networks (ANN), support vector machines (SVM), random forest (RF), regression trees,

deep neural networks (DNN) and extreme gradient boosting (XGBoost) to enhance the prediction models for soil organic carbon (SOC). A genetic algorithm (GA) was employed as a feature selection approach to identify the most effective variables. The results of our study revealed that precipitation emerges as the most influential predictor, accounting for 14.9% of the spatial variability in SOC. These findings provided valuable insights into the key drivers of SOC variability, contributing to our understanding of soil dynamics and carbon sequestration.

*Padmapriya & Sasilatha (2023)* have suggested that in soil classification, utilizing both deep learning and machine learning models has become essential for accurately determining soil types. To this end, a novel multi-stacking ensemble model was introduced in conjunction with a unique feature selection algorithm called Q-HOG. This approach capitalized on the advancements in Artificial Intelligence (AI), particularly in the context of smart agriculture. The method of designing a multi-stacking ensemble for multi-classification leverages both deep learning and machine learning approaches, though at the cost of increased computation time.

*John et al. (2020)* have proposed predictors include Base Saturation (BS), Effective Cation Exchange Capacity (ECEC), Potassium to Magnesium Ratio (K_Mg), Calcium to Magnesium Ratio (Ca_Mg), elevation, total catchment area, topographic wetness index, Ratio Vegetation Index (RVI), Normalized Difference Build-Up Index (NDBI), Normalized Difference Vegetation Index (NDVI), Soil Adjusted Vegetation Index (SAVI), Normalized Difference Moisture Index (NDMI) and Land Surface Temperature (LST).

*Omondiagbe et al. (2023)* have designed a method for building a convolutional neural network (CNN) for soil spectroscopic and evaluating its performance with data from the Kellogg Soil Research Institute's dataset and the LUCAS soil libraries. There were two stages to this strategy. Initially, it mechanized the creation of a network with every link. This involved automatically choosing the various layer types and the number of cells in each layer. The first tactic was to modify the Populations Training (PBT) technique by substituting a Bayesian optimization methodology for the random search method employed in PBT.

*Peng et al. (2023)* have attempted to improve the accuracy forecasts of extremely heterogeneous in the soil of a small-scale manufacturing site by developing efficient 3D forecasting algorithms using artificial intelligence and easily accessible multisource supplementary data. To develop six individual and two collective models for Zn using raw covariates from stratigraphic succession, electrical resistivity tomography, functional area layout, and derived covariates. These models were based on machine learning algorithms like random forest, extreme gradient boosting, and k-nearest neighbors as well as the stacking method employed by ensemble ML.

*Zhao, Wei & Wen (2022)* have designed a model for forecasting the enhanced deep Q network. The model's convergence speed was increased, and the acquisition rate of samples used for training for agents in deep Q networks was accelerated through the reuse of condition values. Simultaneously, a flexible fuzzy membership component was used to modify the agent's responsiveness towards external feedback values over various training intervals and enhance the model's ability to remain stable following completion. In order to

predict observed outcomes for interpolated points, an adaptive inverse length interpolated technique was finally used, which increases the model's predictive accuracy.

*Tripathi, Tiwari & Tiwari (2022)* have proposed estimating wheat crop yield utilizing various data parameters, including optical remote sensing satellite data, soil health parameters, and SAR backscatter. Among the models employed, the Deep Learning Multilayer Perception (DLMLP) model based on soil health performed exceptionally well in estimating crop yield. Notably, when compared to the Ordinary Least Squares Regressor (OLS), the DLMLP test R2 showed a significant improvement of 42.2%. Remarkably, the DLMLP model delivered satisfactory accuracy in yield estimation, even in the absence of historical validation data for soil health parameters in the preceding years for wheat seasons.

### Research gaps and challenges

Numerous advancements and limitations in conventional hydroponic and soil compound prediction during plant growth are listed in Table 1. Gradient boosting regression trees have been regarded as a promising tool to tackle the impact of chemicals on the environment. It is used for image recognition (*Gao et al., 2021*). Root concentration factor in plant-soil system is regarded as the difficult one. Ensemble has the ability to determine the reliability for better prediction. But, it is affected by diverse climate and agro-ecological structures (*Emadi et al., 2020*). The ensemble model is used to validate as well as to detect the kind of clay soil (*Padmapriya & Sasilatha, 2023*). It is also used to determine the fertility of the soil and detect the weed. But it consumes more time for computation. The ensemble model has the ability to provide better classification outcomes as well as random features (*John et al., 2020*). It has permitted precise soil management for crop production. Here, the auxiliary predictors need to be enhanced. CNN and Bayesian optimization have been used to choose the hyperparameters tuning strategy (*Omondiagbe et al., 2023*). It has the ability to determine the optimal data processing phases. But, there is a lack of soil properties. Machine learning has the ability to retrieve features through images. It is used to train the classifier, which differentiates the plants (*Peng et al., 2023*). However, this model acquires more data. Deep Q Network is used to enhance the stability of the model after convergence (*Zhao, Wei & Wen, 2022*). The predicted accuracy of the soil is high in this model. But, it requires more time for the training phase. DLMLP is used to measure the optimum levels of diverse soil health parameters (*Tripathi, Tiwari & Tiwari, 2022*). It is useful for the earlier resolution in soil health parameters. It is failed to determine the crop yield. These challenges help to develop better deep learning-based hydroponic and soil compound prediction.

## AN AUTOMATED MODEL OF HYDROPONIC AND SOIL COMPOUND PREDICTION DURING PLANT GROWTH USING ADAPTIVE TECHNIQUE

This section describes the proposed adaptive technique used for prediction.

**Table 1 Characteristics and defects of existing work hydroponic and soil compound prediction during plant growth.**

| Author (citation) | Methodology | Features | Challenges |
|---|---|---|---|
| *Gao et al. (2021)* | Gradient boosting regression tree | • It has been regarded as the promising tool to tackle the impact of chemicals in the environment.<br>• It is used for image recognition. | • Root concentration factor in plant-soil system is regarded as the difficult one. |
| *Emadi et al. (2020)* | Ensemble | • It has the ability to determine the reliability for better prediction. | • But, it is affected by diverse climate and agro-ecological structures. |
| *Padmapriya & Sasilatha (2023)* | Ensemble | • This model is used to validate as well as to detect the kind of clay soil.<br>• It is also used to find the fertility of the soil and detect the weed. | • But, it consume more time for computation. |
| *John et al. (2020)* | Ensemble | • This model has the ability to provide better classification outcome as well as the random features.<br>• It has permitted precise soil management for crop production. | • Here, the auxiliary predictors are need to be enhanced. |
| *Omondiagbe et al. (2023)* | CNN | • Here, the Bayesian optimization has been used to choose the hyper parameters tuning strategy.<br>• It has the ability to determine the optimal data processing phases. | • But, there is a lack in soil properties. |
| *Peng et al. (2023)* | Machine learning | • It has the ability to retrieve the features through images.<br>• It is used to train the classifier, which differentiate the plants. | • But, this model acquires more number of data. |
| *Zhao, Wei & Wen (2022)* | Deep Q Network | • It is used to enhance the stability of the model after convergence.<br>• The predicted accuracy of the soil is high in this model. | • But, it require more time for training phase. |
| *Tripathi, Tiwari & Tiwari (2022)* | DLMLP | • It is used to measure the optimum levels of diverse soil health parameters.<br>• It is useful for the earlier resolution in soil health parameters. | • It is failed to determine the crop yield. |

## Implemented dataset details

**Dataset (Hydroponic and Soil Compound Dataset):** The data is gathered from the online repository (*High et al., 2019*). The dataset contains measurements of various phytohormone concentrations, which are plant growth hormones. These hormones include zeatin, adenosine, indole-3-acetic acid, abscisic acid, and isopentenyl adenosine. The dataset has been collected to study and understand how the presence of earthworms, different growth, and the concentrations of specific plant growth hormones impact plant growth and related factors. The input data is mentioned by the term $OF_y^{da}$. The data are collected from the standard Kaggle datasets to validate the performance accurately in the hydroponic and soil compound prediction model in the plant growth. However, the entire dataset has been taken for the validation process to split into training and testing process. Here, 75% of data is considered for the training process whereas 25% of the datas is taken in the

testing phase. Due to this, the data quality gets increased in the evaluation process. While collecting the data in the representative and significant way, the potential bias in the dataset gets eliminated to improve the accurate outcomes in the hydroponic and soil compound prediction in plant growth.

## Proposed prediction system and its description

Predicting hydroponic and soil compounds is crucial in agricultural research and practice. It involves modeling and forecasting the concentrations of essential plant growth hormones, assessing the impact of various factors such as growth media and the presence of earthworms, and predicting outcomes related to soil health, hydroponic conditions, and plant biomass. This predictive endeavor has significant implications for optimizing agricultural practices, enhancing crop yield, and promoting sustainable farming techniques. Accurate predictions of hydroponic and soil compounds can lead to optimized crop yield. Predictive models can help in the early detection of issues in plant growth. If the model predicts deviations from expected compound concentrations or plant biomass, it can signal potential problems, allowing for proactive intervention. On the other hand, plant growth is a highly complex process influenced by numerous factors. While predictive models can capture some of this complexity, they may not account for all variables, leading to inaccuracies. The accuracy of predictions heavily relies on the quality of the data used for model training. Incomplete or inaccurate data can lead to unreliable predictions. Addressing these limitations using the proposed model and continually improving predictive techniques is essential for harnessing the full potential of such predictions in agriculture and environmental management. Figure 1 shows the diagrammatic representation of the proposed hydroponic and soil compound prediction model using deep learning.

Our suggested model aims to develop a deep learning-based approach for predicting hydroponic and soil compound dynamics to enable more informed decisions in crop production. Initially, gather the data from the online sources and obtain three features: input feature, weighted feature, and deep feature. In the weighted feature, the weight is optimized using the novel technique named IEMOA to maximize the effectiveness in terms of chi-squared statistics and relief score. To design the MS-CAGRU network, a powerful fusion of multiple deep learning techniques for prediction is utilized. This model combines convolutional autoencoders to capture spatial patterns and encode essential features, GRU for handling sequential and temporal data, which is vital for understanding the dynamics of hydroponic and soil compound systems and multi-scale feature fusion, enabling the model to incorporate information from different levels of granularity. This approach empowers crop growth in both hydroponic and soil-based cultivation systems. Finally, the model's performance is compared with traditional models to demonstrate its superiority. This comparison will underscore the practical benefits of our deep learning-based approach in improving crop production decisions.

## Suggested optimization approach: IEMOA

The major challenges that exist in diverse algorithms for hydroponic and soil compound prediction models in plant growth. Major optimization algorithms are emerged still it lack

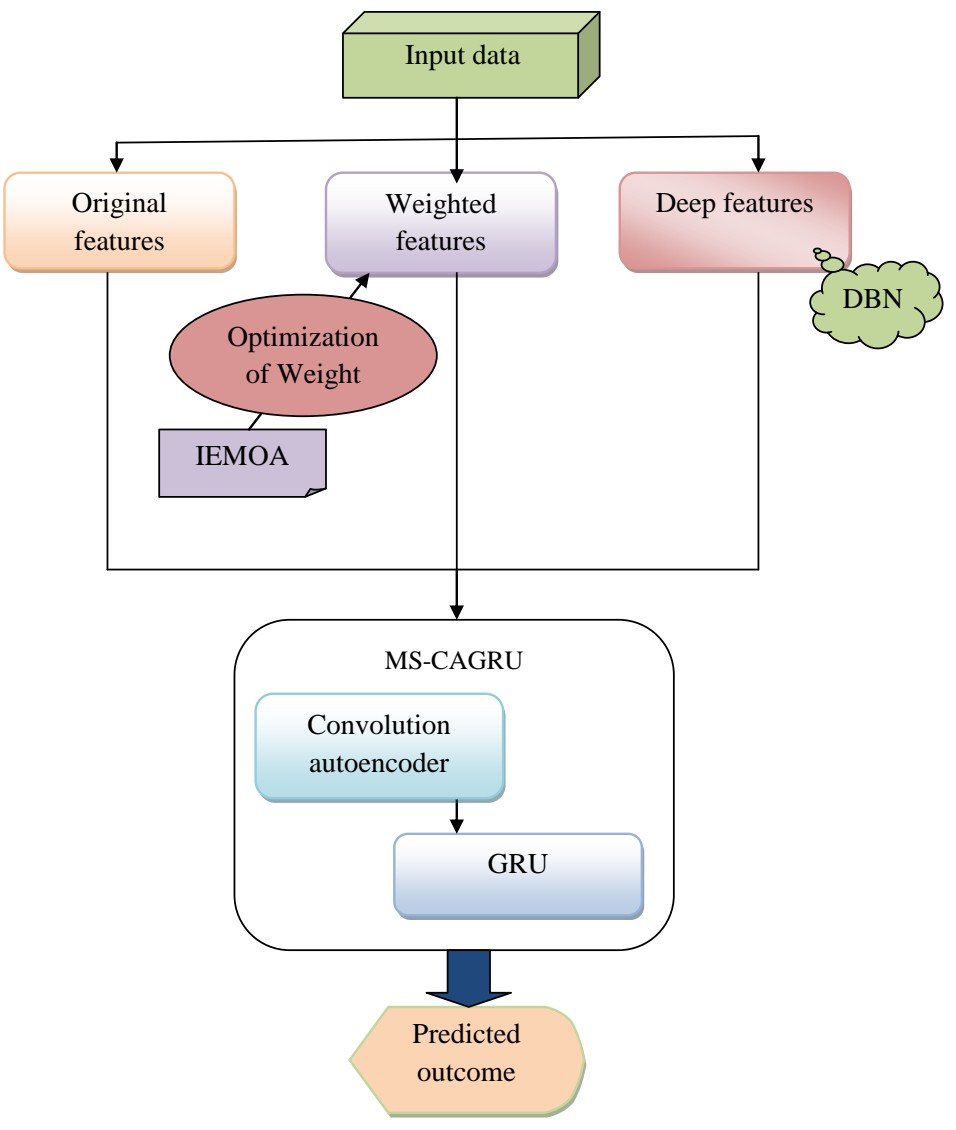

**Figure 1** Delineative representation of the proposed hydroponic and soil compound prediction model using deep learning.

from few challenges that needs to be resolved. While adapting with the larger training samples, often it emerges overfitting or overlapping occurs which tentatively degrades the reliability of the system. Henceforth, the research work adapts an existing Mother Optimization Algorithm (MOA) (*Matoušová et al., 2023*) in order to achieve remarkable precision and faster convergence in comparison to certain metaheuristic algorithms. However, it is worth noting that MOA alone does not provide solutions for problems related to hydro-thermal scheduling and unit commitment in power optimization. In order to address these specific challenges, a novel optimization approach, known as IEMOA, has been introduced to predict the compound of hydroponic and soil systems.

This IEMOA algorithm shows better performance in terms of convergence analysis. Due to this, the overfitting gets reduced to enhance the system performance in hydroponic and soil compound prediction plant growth. Here, a technique is employed where random integers are updated based on the worst and best fitness values. This formulation is mathematically represented in Eq. (1).

$$\lambda = -t * \frac{1}{Max_{iter}}. \tag{1}$$

Here, the random variable is stated as $\lambda$. The variable $Max_{iter}$ defines the maximum iteration, and the variable $t$ symbolizes the current iteration. The random variable is updated in the recommended IEMOA in Eq. (5). The mathematical model of the suggested IEMOA is described below.

The home is unquestionably the first educational institution in society, and a mother's role as an educator is crucial while growing children. A mother imparts valuable life lessons and encounters with life to her children, and they grow as a result of her guidance. The three procedures listed below are some of the most significant ways that a mom and her children engage of (i) training, (ii) guidance, and (iii) nurturing. Thus, the suggested MOA makes advantage of computational modeling of compassionate and instructive actions.

The suggested MOA is a population-level metaheuristic method that uses an iterative approach to tackle optimization issues. The user base of the method is made up of potential fixes that are shown as variables in the issue space. Equation (2) denotes the matrix model of the population as a whole. Set at the beginning of the optimization procedure using Eq. (3).

$$U_i = \begin{bmatrix} U_1 \\ \vdots \\ U_i \\ \vdots \\ U_s \end{bmatrix}_{s \times m} = \begin{bmatrix} U_{1,1} & \cdots & U_{1,d} & \cdots & U_{1,m} \\ \cdots & \ddots & \vdots & \vdots & \vdots \\ UU_{i,1} & \cdots & U_{i,d} & \cdots & U_{i,m} \\ \vdots & \vdots & \vdots & \ddots & \vdots \\ U_{s,1} & \cdots & U_{s,d} & \cdots & U_{s,m} \end{bmatrix}_{s \times m} \tag{2}$$

$$u_{i,} = \gamma_j + \lambda.(\gamma_j - \kappa_i), i = 1, 2 \ldots \ldots s, j = 1, 2 \ldots \ldots m. \tag{3}$$

Here, $s$ is the quantity of a population's participants, $m$ is the number of decisive factors, and $U$ is the population size matrix of the suggested MOA. The variable $\lambda$ provides the random interval which lies in the interval of $(0, 1)$. Also, the upper bound and lower bound are defined as $\gamma_j$ and $\kappa_i$. The term $U_i$ is equal to $U_i = u_{1,i} \ldots u_{1,j} \ldots u_{i,m}$.

All members of the MOA population offer a possible answer to the problem under optimization and the values that each component of the group suggests for the decision variables can be used to calculate the objective value for the issue at hand. The function parameters with objectives can be expressed mathematically as a vector by applying Eq. (4).

$$K = \begin{bmatrix} K_1 \\ \vdots \\ K_i \\ \vdots \\ K_s \end{bmatrix}_{s \times 1} = \begin{bmatrix} K(y_1) \\ \vdots \\ K(y_i) \\ \vdots \\ K(y_S) \end{bmatrix}_{s \times 1} . \tag{4}$$

In the above context, the term $K$ is the final goal function value for the $i$th candidate option and $K_i$ is a collection of unbiased parameter values.

**Phase 1: training (exploration phase):** Applying Eq. (5), another position is produced for every individual during this phase. As Eq. (6) demonstrated, the new positioning is recognized as the appropriate member's position if the objective function improves.

$$U_{i,j}^{p_1} = (u_{i,j}^{p_1} + \lambda_{(0,1)}).(l_j - \lambda(2).u_{i,j}^{p_1}) \tag{5}$$

$$U_i = \begin{cases} \{u_i^P, T_i^P < K_i \\ u_i^P, else \end{cases}. \tag{6}$$

Here, the term $K_{i,j}$ is the $j_{th}$ component of the starting point of the $i_{th}$ demographic participant and $T_i$ is the $j_{th}$ parameter of the mother's orientation. The new standing, $K_{i,j}$, is obtained for the $i_{th}$ populous member using the first within the MOA. The term $\lambda(2)$ defines the random function that uniformly generates the random number at the interval of $(1, 2)$. Here, the variable $\lambda$ symbolizes the random number; in conventional approaches, the random uniform number is produced using the function range of $(0, 1)$. This may lead to optimal issues, and these issues can be mitigated using the proposed formulation in Eq. (1).

**Phase 2: guidance (exploration phase):** Each individual's set of poor conduct $KKl_{i,j}$ isascertained by applying Eq. (7) to compare the value of the objective function for that member. A component is uniformly chosen at random from the created collection of unwelcome actions, $KKl_{i,j}$, for each $u_{i,j}$. Using (8), a new location is initially built for each member to simulate protecting the child from harmful actions. In the event that it raises the value of the goal's function, this additional position takes the place of the related individual's prior role, as determined by Eq. (9).

$$JJ_i = \{u_k, K_y > K_i \in l\{1, 2, \ldots . s\}, where\ i = 1, 2 \ldots . s \tag{7}$$

$$u_{i,j} = u_{i,j} + \lambda.(u_{i,j} - \lambda(2).KKl_{i,j}) \tag{8}$$

$$U_i = \begin{cases} \{u_i^P, K_i^P < K_i \\ u_i^P, else \end{cases}. \tag{9}$$

From the above equation, the term $KKl_{i,j}$ is the chosen unwanted conduct for the $i_{th}$ population participant $KKl_{i,j}$ is its $j_{th}$ dimension, and $u_{i,j}$ is the set of poor behavior for the $i_{th}$ community member.

**Phase 3: Nurturing (exploitation phase):** Mothers encourage their children to develop their talents throughout their schooling in a variety of ways. By slightly altering the population members' positions, parenting increases neighborhood searching and exploitation capacity during the MOA phase. To replicate the period of rising, every member of the community is first given a new position based on the modeling of kids' growth in personalities with the help of Eq. (10). If the value of the goal function increases in the new position, the newly created position is substituted for the prior one held by the matching member, as specified in Eq. (11).

$$u_i^{p1} = u_i^{p1} + \gamma \times (1 - 2\lambda) \frac{\times(\gamma_j - \kappa_i)}{m} \tag{10}$$

$$u_i = \begin{cases} \{U_i^P, K_i^P < y_i \\ U_i^P, else \end{cases}. \tag{11}$$

In the above context, $U_i^P$ is the new location determined by the third stage of the planned MOA, the term $U_i^P$ for the $i_{th}$ populace member. When $i$ is the result of the goal of the function, where $m$ is the repetition counter's truthful value. The pseudocode presented IEMOA is presented in Algorithm 1. Figure 2 shows the recommended IEMOA model.

## EXTRACTING THE FEATURES AND WEIGHTED FEATURES USING DBN AND IEMOA WITH THE OBJECTIVE DERIVATION

This section explains the extracting and weighted features using the proposed method in detail.

### Original and weighted features

In this phase, the collected data is provided as the original feature, which is represented as $OF_y^{da}$. Also, multiplying the original data with these optimized weights generates a set of weighted features represented as $WF_T^{da}$. Optimizing the weights can significantly enhance the performance of machine learning models. It allows the model to assign more importance to relevant features, which can lead to better predictions and reduced overfitting. The weight is optimized by using the proposed IEMOA approach. This step aims to enhance the relevance of certain features, potentially amplifying their impact on subsequent analyses and predictions. Figure 3 shows the view of original and weighted features.

### DBN-based features

A DBN is a type of neural network that contains many layers of interconnected neurons or nodes. It is typically composed of two main types of layers: a stack of Restricted Boltzmann Machine (RBMs) in the lower layers and a layer of traditional neural network neurons in the upper layer.

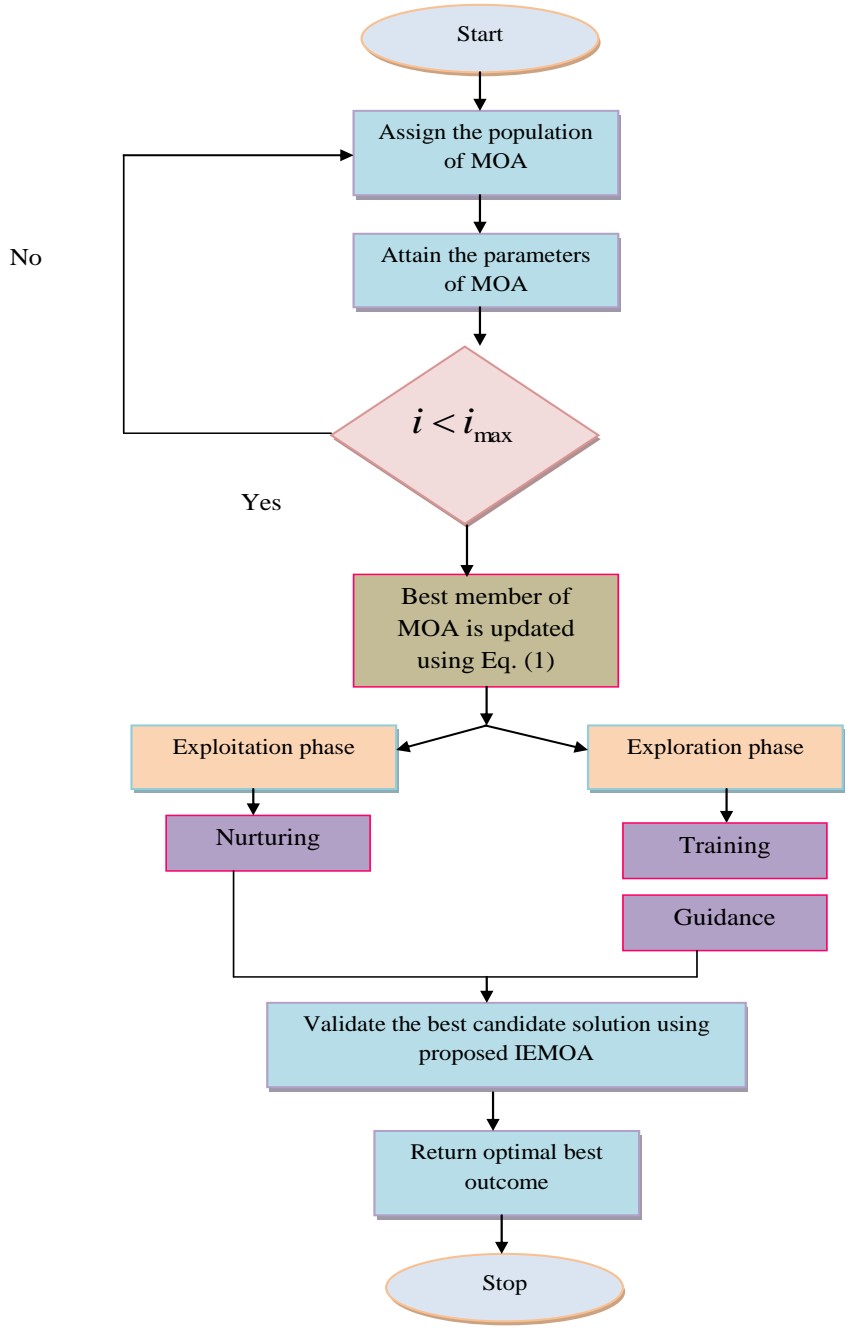

**Figure 2** **Proposed IEMOA flowchart.**

The collected data $RF_y^{da}$ is the input to this phase, to address this limitation; a hierarchical learning system can be employed, wherein the learned features from one RBM are used as input data for a subsequent RBM. This layer-by-layer approach is commonly used to construct a DBN (*Zhao, Zhang & Zheng, 2017*), enabling the progressive extraction of deep features from the input data.

---

**Algorithm 1: IEMOA**

---

**Input:** Optimized weight $W_e$, Relief score $\mu$, Chi-squared statistic $\varphi^2$

**Output:** Optimal solution

Begin MOA algorithm

Determine the number of iterations and size of the population

Generate an initial population of potential solutions based on Eq. (2)

Calculate the fitness function for each population

Initialize $m$

For $m = 1:M$

While $(i < i_{\max})$do

Best random value is updated using Eq. (1)

For $i = 1:S$

**Phase 1: Education**

Update the position of $i_{th}$ members in the population in Eq. (5)

Update the $i_{th}$member's attributes by Eq. (6)

**Phase 2: Guidance**

Compute the updated position of the $i_{th}$ member in the population according to Eq. (7)

Modify the attributes of the $i_{th}$member, such as its position, velocity, or other relevant properties, as prescribed by Eq. (8)

**Phase 3: Nurturing**

Compute the updated position of the $i_{th}$ member in the population according to Eq. (10)

Modify the attributes of the $i_{th}$member, such as its position, velocity, or other relevant properties, as prescribed by Eq. (11)

End

Most promising candidate solution is preserved

Optimal solution is obtained

End

---

DBN is a type of deep learning model used in machine learning and neural network research. They are composed of multiple layers of stochastic, generative neural networks, which include RBMs. The key idea behind DBNs is to learn a hierarchical representation of the input data, with each layer capturing increasingly abstract and high-level features.

A RBM is often employed as a building block in the layer-wise learning of a DBN. It consists of two layers, representing a specific type of Markov random field, with visible unit's denoted as $h = 0, 1^D$ and units of hidden as $h = 0, 1^F$. The energy associated with a joint configuration of these units is given by Eq. (12). The term $g_i$ and $y_i$ defines the bias of the unit.

$$T(h, j, k) = -\sum_{i=1}^{D} g_i, b_i - \sum_{i=1}^{D} y_i, j_i - \sum_{i=1}^{D}\sum_{j=1}^{u} T_i, u_i, y_i. \tag{12}$$

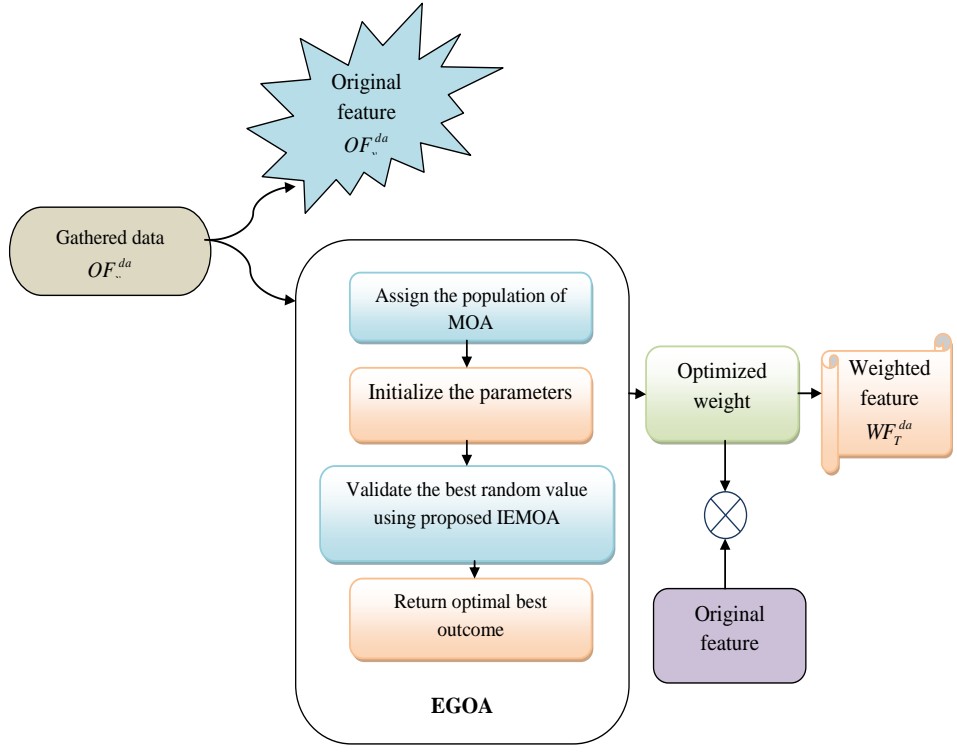

**Figure 3    View of original and weighted features.**

The variable $T(h, j, k)$ collectively defines the model's characteristics and plays a crucial role in determining the energy of a configuration and, consequently, the probability distribution associated with the RBM and, mathematically shown in Eqs. (13) and (14).

$$T(h, j, k) = \frac{1}{Z(\theta)} \exp(-T(h, j, k)) \tag{13}$$

$$Z(\theta) = \sum_v \sum_h (-T(h, j, k)). \tag{14}$$

Hence, the allocating function is defined as $Z(\theta)$. Once the hidden units are selected, reconstruct the input data by setting the probability of Eq. (14). The strength of RBMs is rooted in their reconstruction-oriented training approach. In the reconstruction process, RBMs rely solely on the information contained within their hidden units, which have been trained to capture meaningful features from the input data. Finally, get the deep features, which are represented as $DF_y^{da}$. Figure 4 shows the architecture diagram of DBN-based features.

## Fitness equation for weighted features

The three different types of features offer a diverse and comprehensive set of attributes for subsequent analyses. Optimizing the weights is a crucial step in various data analysis

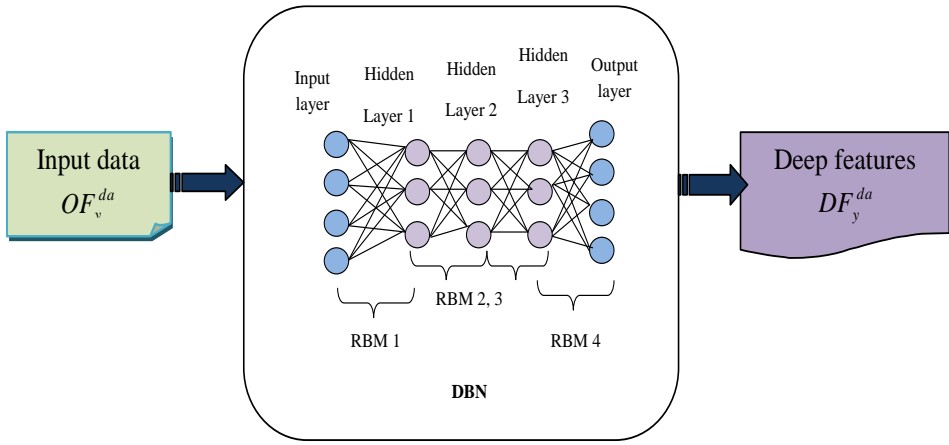

**Figure 4** Architecture diagram of DBN based features.

and modeling tasks. Feature weight optimization can make models more interpretable. Emphasizing certain features can lead to a more understandable model, making it easier to explain the factors driving the predictions or decisions. When feature weights are not optimized, all features are treated equally in the analysis, regardless of their relevance or importance. This can lead to suboptimal results, especially if some features are more informative than others. To lessen these kinds of drawbacks optimized weight parameters, with the help of the suggested IEMOA approach. The objective formulation of the weighted feature is shown in Eq. (15).

$$TL_1 = \arg \max_{\{W_e\}} (\phi^2 + \mu).$$  (15)

From the above context, the term $\phi^2$ symbolizes the Chi-squared statistic and $\mu$ defines the relief score. Also, the variable $W_e$ shows the optimized weight, and the range is 0.01 - 0.99. The definition and formulation of $\phi^2$ and $\mu$ is specified in Eqs. (16) and (17).

**Chi-squared statistic:** It quantifies the difference between the expected and observed frequencies of each category in the feature concerning the target variable.

$$\phi^2 = chi^2 = \frac{\sum (oij - Eij)^2}{Eij}.$$  (16)

**Relief score:** It measures the relevance of individual features in distinguishing between different classes in a dataset. It evaluates how well a feature can discriminate between instances of the same class and instances of different classes.

$$\mu = u_i - (y_i - NH_i) + (y_i - NM_i).$$  (17)

From the above context, the term $oij$ and $Eij$ denotes the observed and the expected frequency. Moreover, the term $NH_i$ defines the same class and $NM_i$ defines the different classes.

## MULTISCALE FEATURE FUSION-BASED CONVOLUTIONAL AUTOENCODER WITH GRU FOR PREDICTION

This section explains the details of the proposed convolutional autoencoder with GRU for prediction.

### Convolutional autoencoder

A typical conventional autoencoder consists of two layers corresponding to an encoder function denoted as $kw$ anda decoder function symbolized as $ji$ (*Guo et al., 2017*). The primary objective of an autoencoder is to generate a code or representation for each input sample by reducing the MSE between the input data and the corresponding reconstructed output across all the samples, which is mathematically shown in Eq. (18).

$$\min \frac{1}{m} \sum_{m=1}^{m} \left\| ji(kw(y_i)) - y_i \right\|^2. \tag{18}$$

The fully connected autoencoder is defined as Eq. (19).

$$ji(y) = \sigma(w_y) = k$$
$$kw(k) = \sigma(uk). \tag{19}$$

In the context of this model, $y$ and $k$ are both vectors, and $\sigma$ represents an activation function such as sigmoid or ReLu. Once the autoencoder is trained, the resulting code $k$ becomes a new and more meaningful representation of the sample input. This $k$ can subsequently be input into the next autoencoder to create what is known as Stacked Autoencoders (SAE). To leverage the data, a convolution autoencoder is defined as shown in Eq. (20).

$$ji(y) = \sigma(y * w) = k$$
$$kw(k) = \sigma(h * u). \tag{20}$$

From the above equation, the term $k$ and $y$ are tensors and also the term $*$ convolution operator. Figure 5 shows the architecture diagram of the convolution autoencoder.

### Gated recurrent unit (GRU)

The GRU is a network that builds upon the long short-term memory (LSTM) architecture, with the goal of improving the efficiency of the LSTM structure while maintaining its efficiency (*Li et al., 2021*). When compared to the LSTM network, the GRU network simplifies the structure by having only two gate mechanisms: the update gate and the reset gate. These gates play a crucial role in addressing prediction challenges in time series data with long intervals and delays. The update gate regulates how much information from the previous time step is incorporated into the current time step, allowing for flexible control over the memory of the network. In contrast, the reset gate determines the degree to which the network should ignore information from the previous time step, allowing for selective retention of relevant information.

The gate of reset output at a time is denoted as $k_t$, and the gate of update output at a time is represented as $y_t$. Additionally, $l_t$ and $l_{t-1}$ correspond to the outputs at time $t$ and

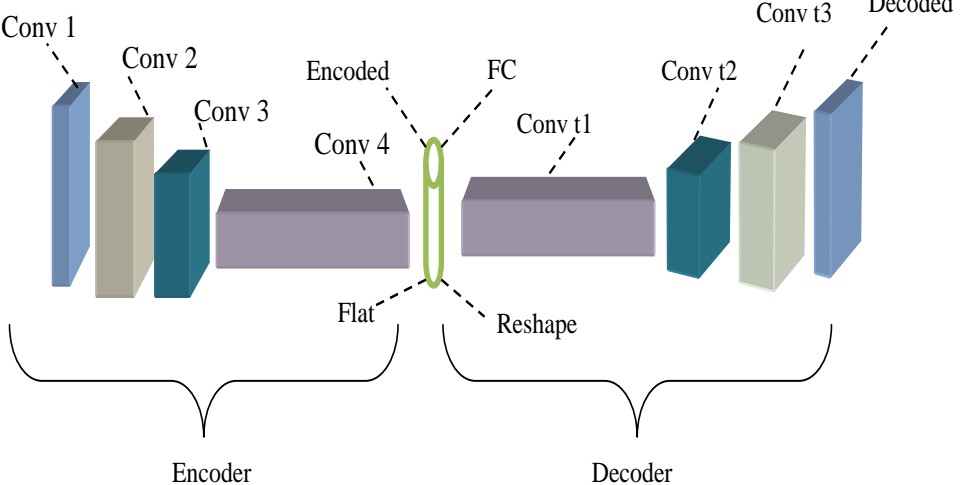

**Convolution Autoencoder**

**Figure 5** Architecture diagram of convolution autoencoder.

$l_{t-1}$, respectively. The term $y_t$ signifies the input at time $t$, and $k_t$ represents the function of activation. The computation process for the memory unit can be described using Eqs. (21) to (24).

$$k_t = \partial(w_r.[l_{t-1}, y_t]) \tag{21}$$

$$y_t = \partial(w_r.[l_{t-1}, y_t]) \tag{22}$$

$$\hat{l}_t = \tan l(w_r.[k_t * l_{t-1}, y_t]) \tag{23}$$

$$l_t = (1 - h_t) * l_{t-1} + y_t. \tag{24}$$

The activation function is termed as $\partial$. From the above equations, the term [] defines the vector representation of the two product matrices, and also the matrix product is defined as $*$. Figure 6 shows the architecture diagram of GRU.

## Proposed MS-CAGRU for prediction

The MS-CAGRU network model represents an innovative and powerful solution for hydroponic and soil compound prediction. By combining convolutional autoencoders, GRU, and multi-scale feature fusion, this model empowers crop growth in both hydroponic and soil-based systems. Thus, the developed technique has the ability to handle the complex datas while the training process takes place. Moreover, the computational complexity of

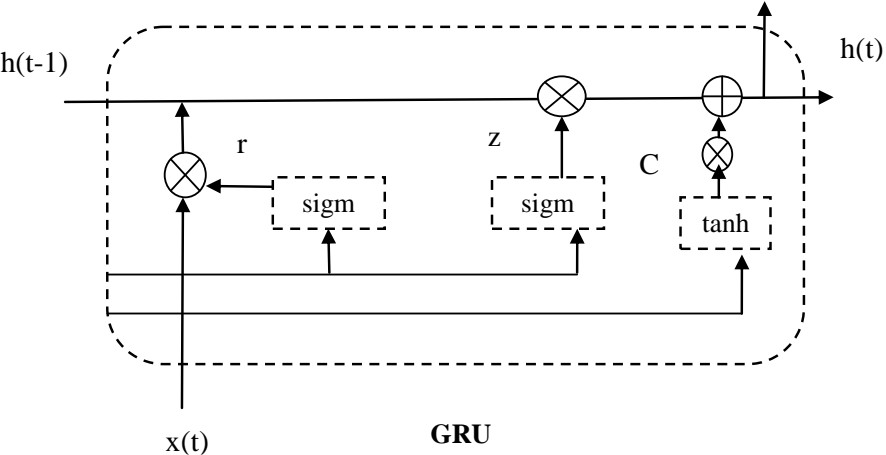

h(t-1)                                      h(t)

x(t)               **GRU**

**Figure 6**   **Architecture diagram of GRU.**

the developed MS-CAGRU model offers better performance in the hydroponic and soil compound prediction model. Due to this, the recommended model provides accurate outcomes the model offers more stable and reliable performance than the existing approaches.

**Multiscale:** This network leverages multi-scale feature fusion techniques, enabling the model to cover the data at different levels of granularity. This approach enhances the model's ability to adapt to various aspects of the data. In our proposed model, the original feature $OF_y^{da}$, weighted features $WF_T^{da}$, and deep features $DF_y^{da}$ are given as input to each scale of the encoder side of the convolution autoencoder, finally these features are combined that is denoted as $EF_y$. This feature extraction aims to extract the most valuable information from the data, enhancing the overall quality of the analysis and insights obtained.

**MS-CAGRU:** The integration of the extracted features $EF_y$ as input to the next layer of convolution autoencoder and subsequently using the output of the convolution autoencoder as input to the GRU, which is the replacement of the FC (fully connected) layer and finally obtained the predicted outcome. This sequential flow of data through the convolution autoencoder and GRU components offered a comprehensive approach to understanding and predicting hydroponic and soil compound levels. Figure 7 shows the proposed view of MS-CAGRU based hydroponic and soil compound prediction.

## SIMULATION FINDINGS AND DISCUSSIONS

This section discusses the obtained results.

### Experimental setup

This paper employed the Python platform to develop a predictive model for hydroponic and soil compounds using deep learning techniques. To validate the effectiveness of the suggested IEMOA-based prediction method, the results were compared with other classification and optimization techniques. The implementation process involved a

**Encoder of Convolution Autoencoder**

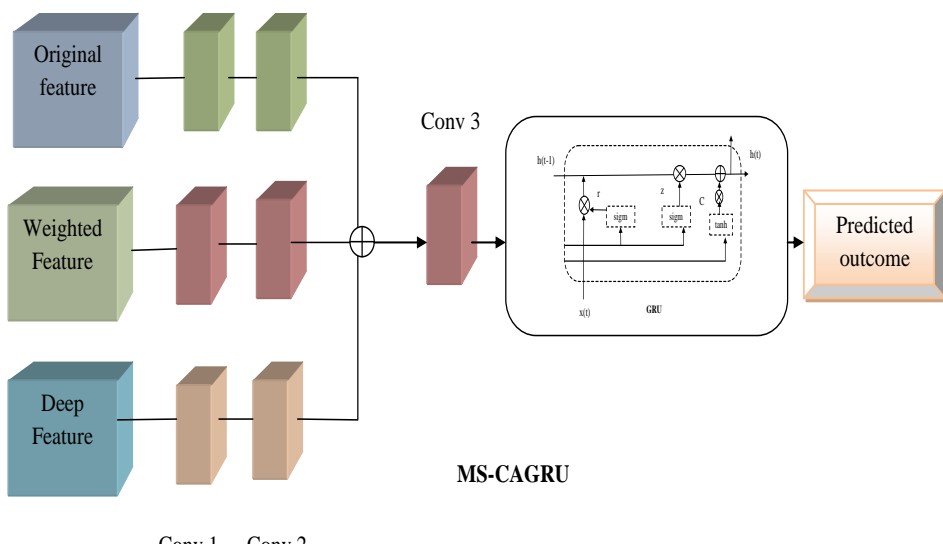

**Figure 7 Proposed view of MS-CAGRU based hydroponic and soil compound prediction.**

population size, chromosome length, and total iterations set at 10, no of features in data, and 50. Various classifiers, including "VGG16 (*Thepade et al., 2022*), InceptionNet (*Zhong & Pun, 2020*), Resnet (*Zhu et al., 2022*), and GRU (*Li et al., 2021*), were utilized to assess the performance of the MS-CAGRU model". Additionally, heuristic approaches such as Beluga Whale Optimization (BWO) (*Gao et al., 2023*), Cheetah Optimizer (CO) (*Akbari et al., 2022*), Squid Game Optimizer (SGO) (*Azizi et al., 2023*), and MOA (*Matoušová et al., 2023*) were employed to validate the efficacy of the suggested IEMOA method. Also, baseline methods like YOLO-EfficientNet (*Wang, Wu & Shen, 2024*), Multiple Linear Regression (MLR) (*Yang et al., 2023*), and Bidirectional Recurrent Neural Network (BRNN) (*Yu et al., 2023*) are computed using the developed model for attaining efficient performance. The basic reporting of the experimental design is listed below. Moreover, the processor of Intel core i3 is used for the implementation with the RAM size of about 8 GB. Hence, the language of phycharm is performed in this research work.

## Performance criteria

Several performance metrics were employed and computed using the following equations to assess the effectiveness of the studied IEMOA model for predicting hydroponic and soil compound outcomes. Due to these diverse performance measures, the evaluation metrics were obtained to prove its efficient performance in the developed model.

(a) The "F1-score" $U_m$ defines the test is conducted to show the accuracy of the model which is estimated using Eq. (25).

$$\text{Um} = \frac{2*qq}{2*(qq+vv+ww)} \tag{25}$$

(b) The NPV, $W_k$ is the measurement of all possible outcomes is measured in Eq. (26).

$$Wk = \frac{xx}{xx + ww} \qquad (26)$$

(c) The "specificity" $W_c$ is defined as the proportion of all negative values which is correctly estimated using Eq. (27).

$$Wc = \frac{xx}{xx + vv} \qquad (27)$$

(d) The computation of "accuracy" $A_i$ is the measurement of accurately predicted outcomes in the hydroponic and soil compound which is estimated using Eq. (28).

$$Ai = \frac{xx + qq}{xx + vv + qq + ww} \qquad (28)$$

(e) The computation of "sensitivity" is the proportion of all positive values that are correctly estimated using Eq. (29).

$$Sn = \frac{qq}{qq + ww} \qquad (29)$$

(f) The false discovery rate (FDR) $Rh$ is the sum of all the rejected outcomes in the number of both false positives and true positives which is estimated in Eq. (30).

$$Rh = \frac{qq}{qq + vv} \qquad (30)$$

(g) The computation of false positive rate (FPR) $Rb$ isthe ratio of analyzing the negative events as well as the number of actual values is evaluated using Eq. (31).

$$Rb = \frac{vv}{xx + vv} \qquad (31)$$

(h) The computation of true positive rate (TPR) $Tb$ isused to evaluate the proportion of the positive predicted samples that is correctly estimated which is expressed in Eq. (32).

$$Tb = \frac{ww}{xx + qq} \qquad (32)$$

(i) The computation of true negative rate (TNR) $TN$ isdefined as the actually predicted negative values is estimated using Eq. (33).

$$TN = \frac{ww}{xx + vv} \qquad (33)$$

(j) The computation of positive predicted value (PPV) or Precision $PPV$ isdefined as the measurement of predicting the positive samples which is computed in Eq. (34).

$$ppv = \frac{ww}{ww + vv} \qquad (34)$$

Here, variable $ww$ and $xx$, $qq$ and $vv$ symbolize true, false positives and false, true negatives.

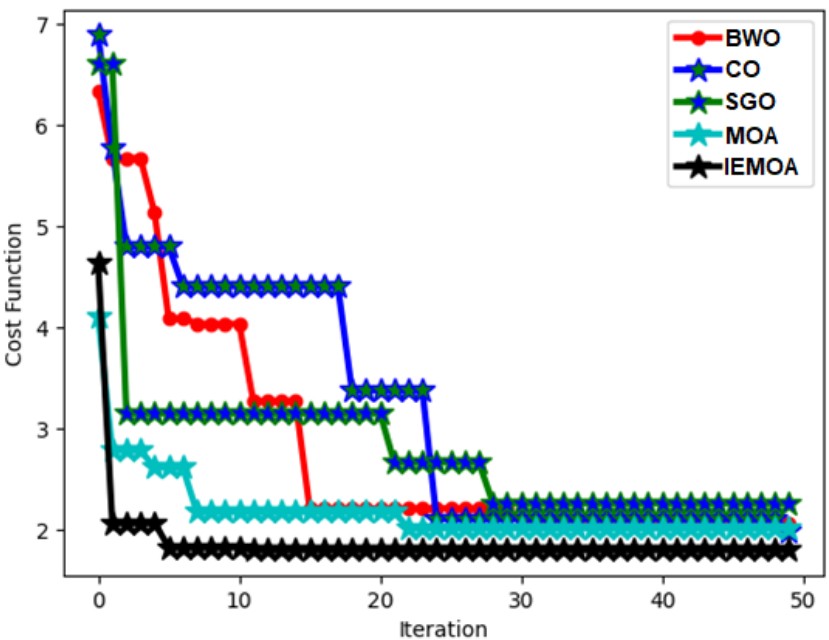

**Figure 8  ROC assessment of the designed hydroponic and soil compounds prediction model compared with various classifiers.**

## Convergence evaluation of the implemented algorithm on hydroponic and soil compounds framework

The assessment of the cost function for the executed approach within the hydroponic and soil compounds prediction framework is depicted in Fig. 8. This illustration demonstrates its performance across different iteration counts. Notably, the cost function of the IEMOA-based hydroponic and soil compounds prediction method is significantly lower than 42.07%, 73.19%, 42.17%, and 51.25% of BWO, CO, SGO, and MOA algorithms, respectively, at the 10th iteration. This analysis underscores the enhanced performance of the IEMOA-based approach.

## ROC assessment of the designed hydroponic and soil compounds prediction model

Figure 9 depicts the ROC assessment of the hydroponic and soil compounds prediction model that was developed. This validation provides insights into the effectiveness of the MS-CAGRU system in predicting hydroponic and soil compounds. The TPR of the MS-CAGRU-based hydroponic and soil compounds prediction system outperforms higher values than VGG16, InceptionNet, Resnet, and GRU by 3.85%, 12.5%, 19.12%, and 30.65%, respectively. This difference in TPR demonstrates the superior predictive performance of the MS-CAGRU-based model for hydroponic and soil compound prediction.

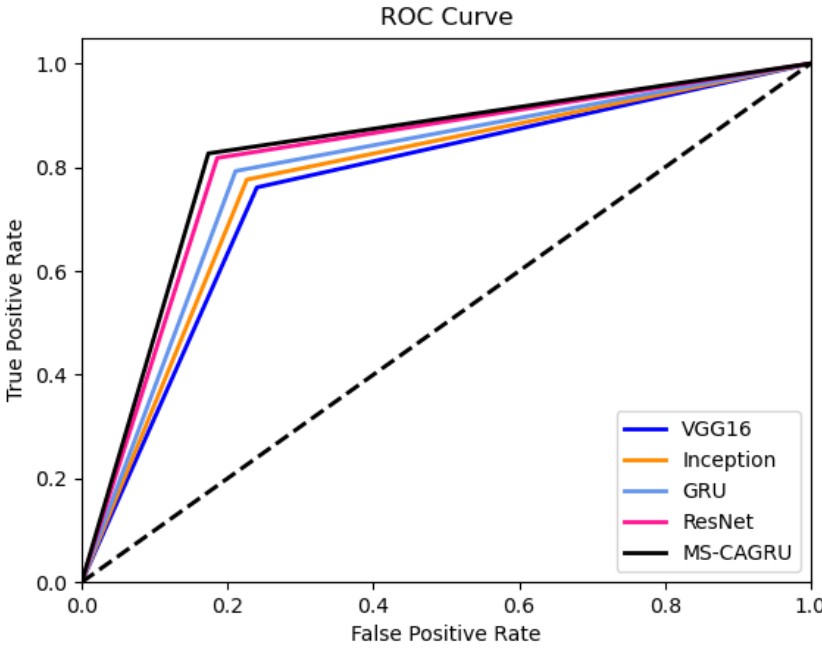

**Figure 9** **Convergence evaluation of the implemented IEMOA approach compared with different algorithms.**

## Validation of the proposed hydroponic and soil compounds prediction model

In Figs. 10, 11, and 12, the validation of the suggested hydroponic and soil compounds prediction model is presented, with a comparison to existing classifiers. In Fig. 10A, the accuracy of the MS-CAGRU-based hydroponic and soil compounds prediction model during the 65th learning percentage, it is observed that it performs 11.11%, 38.46%, 8.57%, and 47.96% more effectively than VGG16, InceptionNet, Resnet, and GRU, respectively. This outcome establishes that the proposed MS-CAGRU-based hydroponic and soil compounds prediction model framework delivers superior performance when compared to other existing models.

## Performance analysis of the suggested hydroponic and soil compounds prediction model

The performance analysis of our recommended hydroponic and soil compounds prediction model is presented in Table 2, which shows the comparison with the conventional classifiers. The MS-CAGRU-based model framework demonstrates superior performance compared to VGG16, InceptionNet, ResNet, and GRU models by 7.2%, 2.95%, 5.18%, and 1.05%, respectively. This outcome establishes that the proposed MS-CAGRU-based hydroponic and soil compounds prediction model framework delivers superior performance when compared to other existing models.

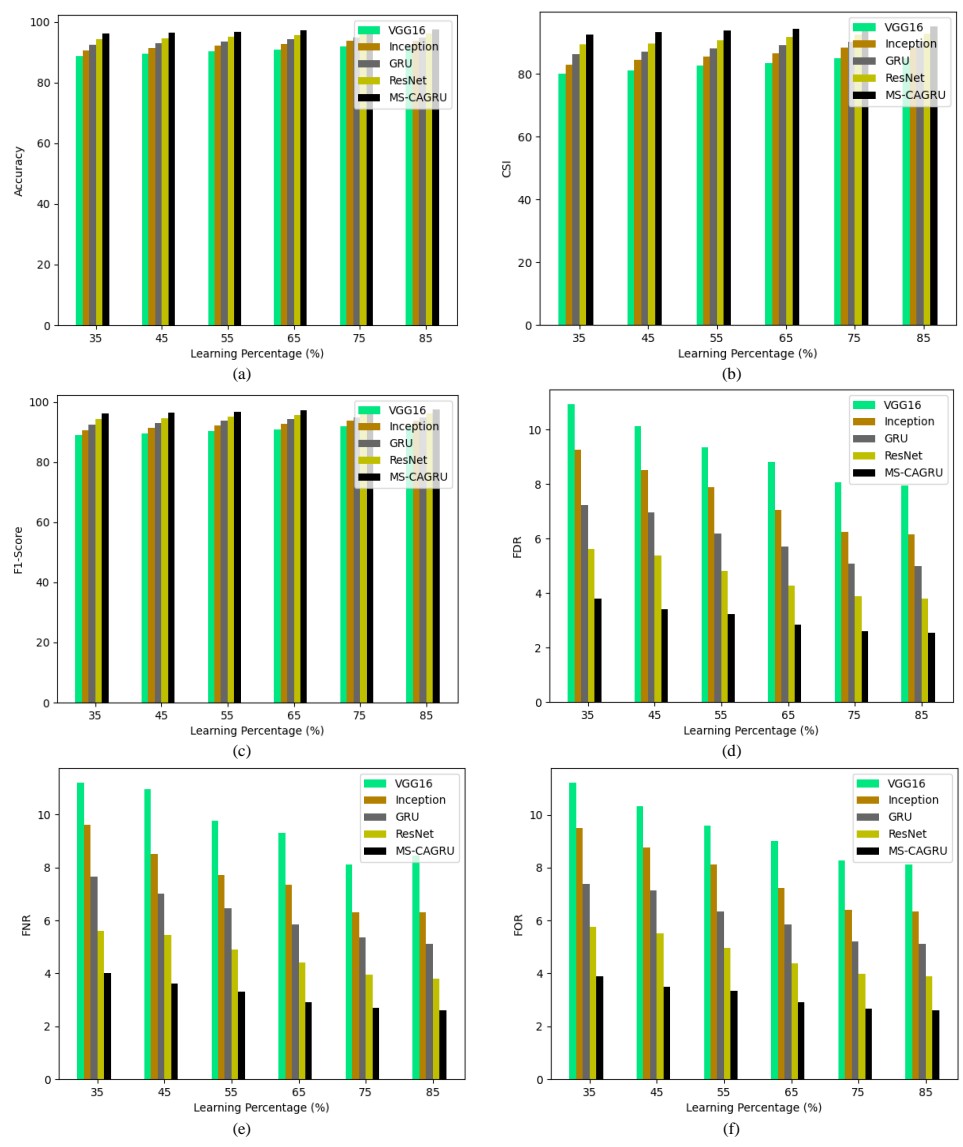

**Figure 10  Performance analysis of the proposed hydroponic and soil compounds prediction model against existing classifiers regarding (A) accuracy, (B) CSI, (C) F1-score, (D) FDR, (E) FNR, (F) FOR.**

## Comparative analysis of the developed model using benchmark methods

The comparative analysis of the developed model for hydroponic and soil compound prediction model is compared with benchmark methods which are shown in Table 3. The developed model shows 3.2%, 2.1%, 1.0%, and 10.6% enhanced performance than YOLO-EfficientNet, MLR, and BRNN in terms of precision measure. Throughout the analysis, the developed model shows enriched performance than the baseline methods.

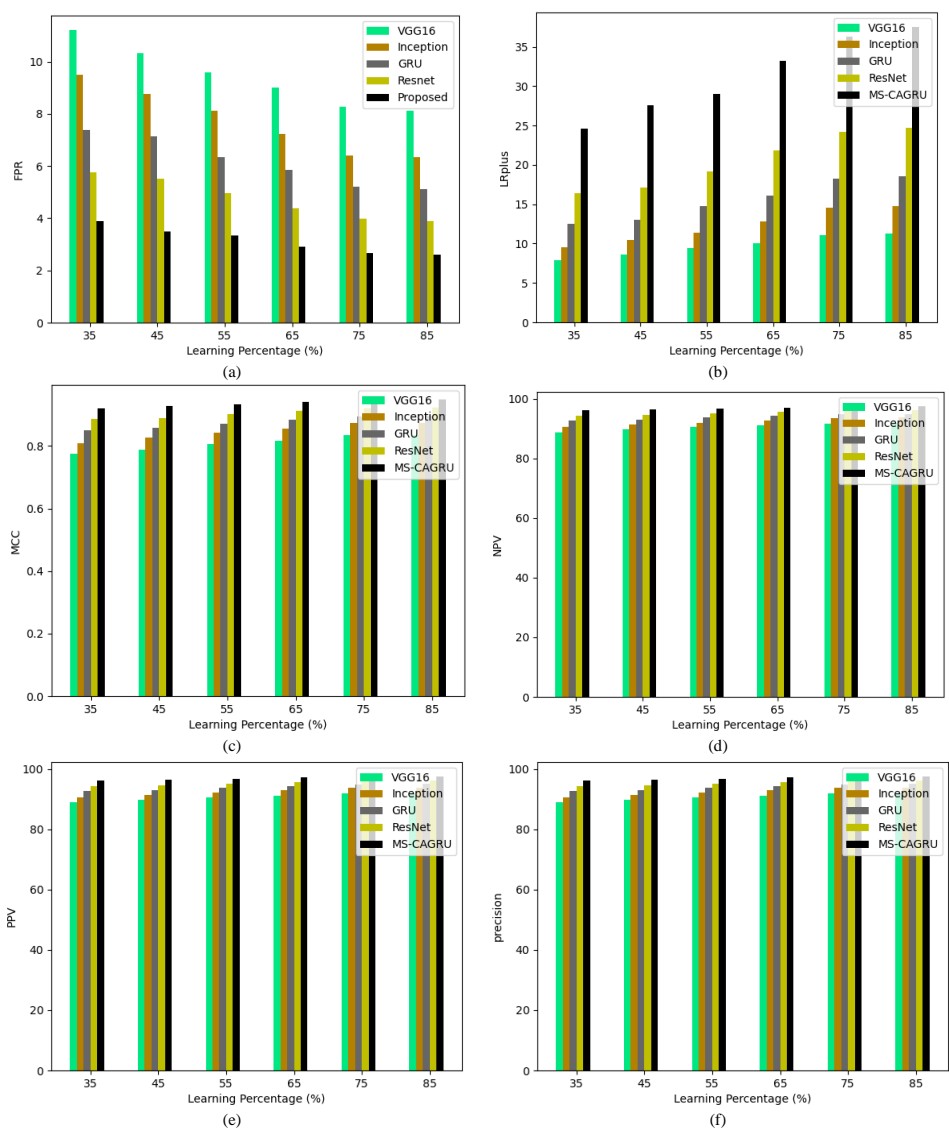

**Figure 11** Performance analysis of the proposed hydroponic and soil compounds prediction model against existing classifiers regarding (A) FPR, (B) LRplus, (C) MCC (D) NPV (E) PPV, (F) Precision.

## Generalization capabilities and scalability of the developed model

The performance of the scalability of the developed model is shown in Fig. 13. Here, the scalability of the developed model is analyzed and varied based on different data sizes in terms of accuracy. Moreover, the developed model offers better scalable performance while handling the different sizes of data to enhance the system performance.

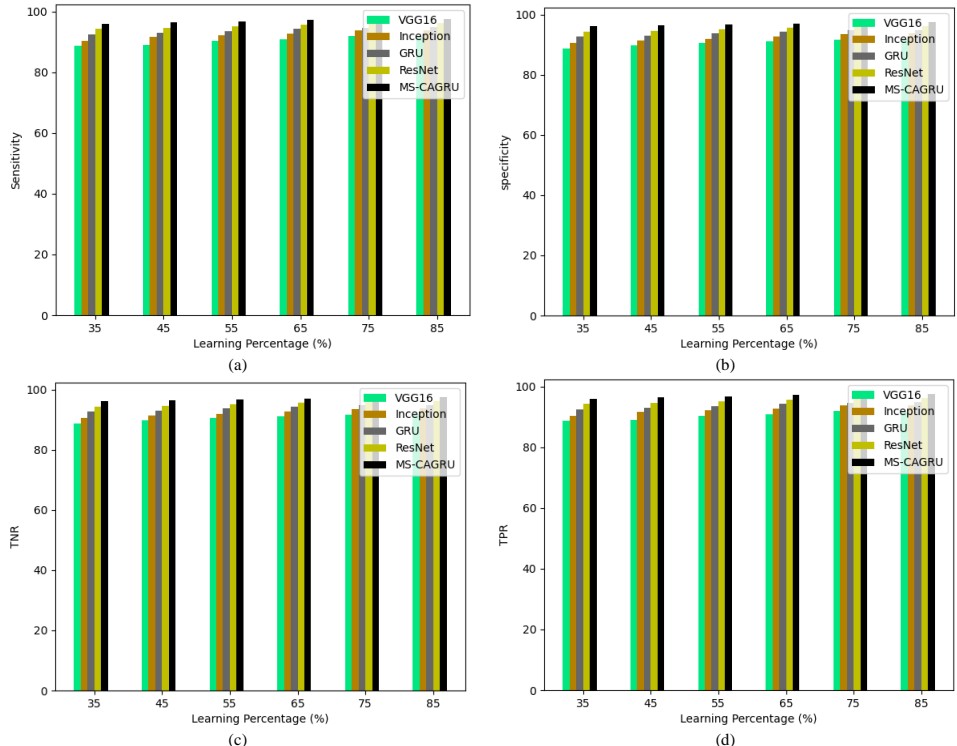

**Figure 12** Performance analysis of the proposed hydroponic and soil compounds prediction model against existing classifiers regarding, (A) Sensitivity, (B) Specificity, (C) TNR, (D) TPR.

## Analysis based on the robustness of the developed model

The robustness analysis is conducted based on the developed model is visualized in Fig. 14. Hence, the robustness analysis is evaluated to prove an accurate outcome in the hydroponic and soil compound prediction model.

# DISCUSSION

## Advantages and disadvantages of deep learning based approaches

The deep learning based approaches have evolved in recent times for hydroponic and soil compound prediction framework which is necessary to plant growth. Recently, numerous deep learning techniques are adapted to increase the productivity based on the plant growth. In traditional approaches, the larger datas becomes complicated to degrade the system performance. While comparing with the traditional approaches, the deep learning model shows superior performance in order to enhance system performance in order to handle complex larger datas while training samples.

## Findings of the results

The resultant analysis shows the accurate findings of the developed model which is described below. Here, the analysis of the developed model offers better performance in the convergence analysis which has the greater capability to minimize the overfitting issues

**Table 2  Characteristics and defects of existing work hydroponic and soil compound prediction during plant growth.**

| Metrics/Classifiers | VGG16 (*Thepade et al., 2022*) | Inception-Net (*Zhong & Pun, 2020*) | ResNet (*Zhu et al., 2022*) | GRU (*Li et al., 2021*) | MS-CAGRU |
|---|---|---|---|---|---|
| Accuracy | 91.8 | 93.64 | 94.72 | 96.04 | 97.32 |
| Sensitivity | 91.87697 | 93.69085 | 94.63722397 | 96.05678 | 97.31861 |
| specificity | 91.72078 | 93.58766 | 94.80519481 | 96.02273 | 97.32143 |
| precision | 91.94949 | 93.7648 | 94.93670886 | 96.1326 | 97.39542 |
| FPR | 8.279221 | 6.412338 | 5.194805195 | 3.977273 | 2.678571 |
| FNR | 8.123028 | 6.309148 | 5.362776025 | 3.943218 | 2.681388 |
| NPV | 91.72078 | 93.58766 | 94.80519481 | 96.02273 | 97.32143 |
| FDR | 8.050513 | 6.235201 | 5.063291139 | 3.867403 | 2.604578 |
| F1-Score | 91.91321 | 93.72781 | 94.78672986 | 96.09467 | 97.357 |
| MCC | 0.835968 | 0.872775 | 0.894387547 | 0.920785 | 0.94639 |
| PR | 91.87697 | 93.69085 | 94.63722397 | 96.05678 | 97.31861 |
| TNR | 91.72078 | 93.58766 | 94.80519481 | 96.02273 | 97.32143 |
| PPV | 91.94949 | 93.7648 | 94.93670886 | 96.1326 | 97.39542 |
| FOR | 8.279221 | 6.412338 | 5.194805195 | 3.977273 | 2.678571 |
| LRplus | 11.0973 | 14.61103 | 18.21766562 | 24.15142 | 36.33228 |
| CSI | 85.0365 | 88.19599 | 90.09009009 | 92.48292 | 94.85012 |

**Table 3  State-of-art-Method for the hydroponic and soil compound prediction approach.**

| Metrics/ Classifiers | YOLO-EfficientNet (*Wang, Wu & Shen, 2024*) | MLR (*Yang et al., 2023*) | BRNN (*Yu et al., 2023*) | MS-CAGRU |
|---|---|---|---|---|
| Accuracy | 94.5 | 95.4 | 96.35 | 97.32 |
| Sensitivity | 94.68 | 95.49 | 96.29 | 97.31861 |
| Specificity | 94.32 | 95.31 | 96.41 | 97.32143 |
| Precision | 94.31 | 95.30 | 96.39 | 97.39542 |
| FPR | 5.68 | 4.69 | 3.59 | 2.678571 |
| FNR | 5.32 | 4.51 | 3.71 | 2.681388 |
| NPV | 94.69 | 95.50 | 96.31 | 97.32143 |
| FDR | 5.69 | 4.70 | 3.61 | 2.604578 |
| F1-Score | 94.49 | 95.39 | 96.34 | 97.357 |
| MCC | 89.00 | 90.80 | 92.70 | 94.639 |

to strengthen the developed model in hydroponic and soil compound prediction model for plant growth. While considering the diverse evaluation measures, the developed model gets validated and analyzed with the standard baseline methods. These analyses are prone to prove the superiority of the developed model over the existing approaches. Henceforth, the developed MS-CAGRU model offers 97% which shows better performance. Hence, the prediction of hydroponic is the significant factor where the maintenance and watering the plants gets decreased to increasing the plant growth in a significant approach. Due to its efficient outcome, the developed model has the ability to strengthen the performance in the hydroponic and soil compound prediction approach for the plant growth to provide better yield productivity to the farmers. In terms, the existing model VGG16 model shows

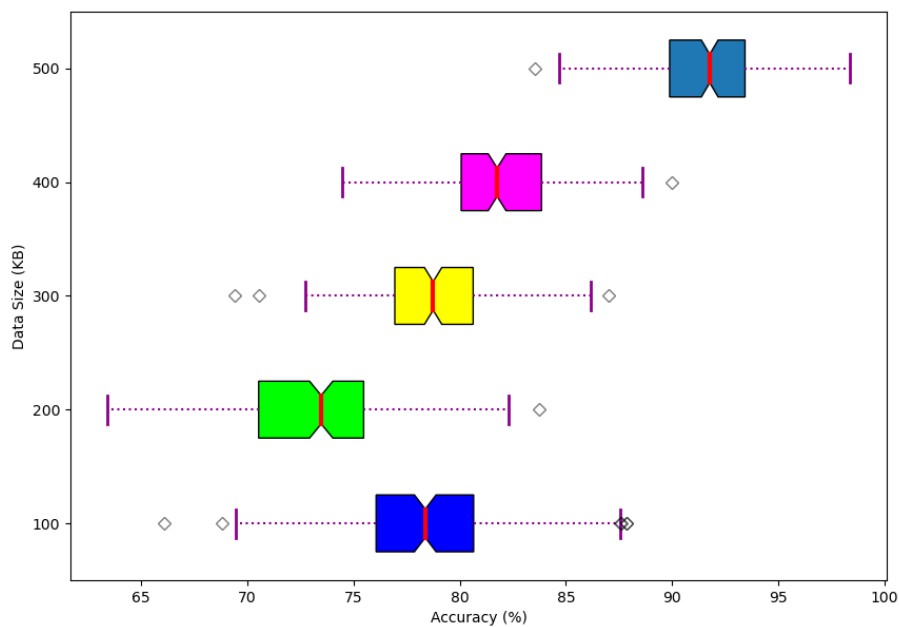

**Figure 13**  **Scalability of the developed model.**

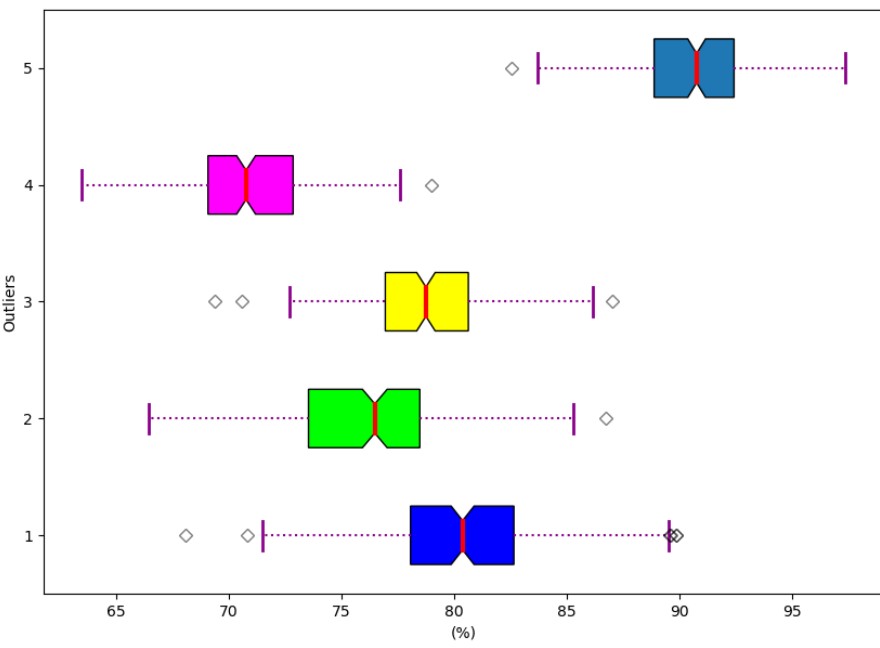

**Figure 14**  **Robustness of the developed model.**

91% which shows lower performance while validating with accuracy analysis. Also, the error rate of FDR analysis of the VGG16 model achieves 8.0% whereas the developed MS-CAGRU model shows 2.6% it significantly reduces the issues of misclassification to provide the accurate predicted outcomes in the plant growth approach.

## CONCLUSIONS

A deep learning-based approach was developed to predict hydroponic and soil compound dynamics, aiming to enhance decision-making in crop production. Initially, data was collected from online sources. The endeavor began with the collection of relevant data from online sources. Three crucial features were identified: input, weighted, and deep features. The weights were optimized using the IEMOA technique, which focused on maximizing the chi-squared statistic and relief score, ensuring the effectiveness of the model. The new model was MS-CAGRU network, a robust fusion of convolutional autoencoders, GRU, and multi-scale feature fusion. This innovative approach empowered crop growth in both hydroponic and soil-based cultivation systems. Following the model's design, its performance was rigorously compared to that of traditional models. This comprehensive comparison highlighted the practical benefits of the deep learning-based approach in improving crop production decisions. When assessing the precision of the MS-CAGRU-based hydroponic and soil compounds prediction model during the 75th learning percentage, it was observed that it performs 21.11%, 58.46%, 81.57%, and 37.96% more effectively than VGG16, InceptionNet, Resnet, and GRU, respectively. This outcome establishes that the proposed MS-CAGRU-based hydroponic and soil compounds prediction model framework delivers superior performance when compared to other existing models. The model heavily relies on the quality and quantity of data collected from online sources. Inaccurate or incomplete data may lead to suboptimal predictions. The proposed hydroponic and soil compound prediction model is a promising advancement in crop production decision-making. However, it is important to be aware of its limitations and address them effectively to ensure its practical and widespread utility in real-world agricultural settings.

### Funding

This work was supported by the Deputyship for Research & Innovation, Ministry of Education in Saudi Arabia through the project number (IFKSUDR_F115). The funders had no role in study design, data collection and analysis, decision to publish, or preparation of the manuscript.

### Grant Disclosures

The following grant information was disclosed by the authors:
The Deputyship for Research & Innovation, Ministry of Education in Saudi Arabia: IFKSUDR_F115.

## Competing Interests

The authors declare there are no competing interests.

## Author Contributions

- Mustufa Haider Abidi conceived and designed the experiments, performed the experiments, analyzed the data, performed the computation work, prepared figures and/or tables, authored or reviewed drafts of the article, and approved the final draft.
- Sanjay Chintakindi conceived and designed the experiments, analyzed the data, authored or reviewed drafts of the article, and approved the final draft.
- Ateekh Ur Rehman conceived and designed the experiments, prepared figures and/or tables, authored or reviewed drafts of the article, and approved the final draft.
- Muneer Khan Mohammed conceived and designed the experiments, performed the experiments, prepared figures and/or tables, and approved the final draft.

## Data Availability

The raw data and code is available in the Supplementary Files.

The Plant growth: Hydroponic and Soil Compound Dataset is available at Kaggle: https://www.kaggle.com/datasets/abtabm/plant-growthhydroponics-and-soil-compound-dataset.

## Supplemental Information

Supplemental information for this article can be found online at http://dx.doi.org/10.7717/peerj-cs.2101#supplemental-information.

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
