# Peer review of "Performance enhancement in hydroponic and soil compound prediction by deep learning techniques"

_PeerJ Computer Science, doi:10.7717/peerj-cs.2101_

## Round 0.1 · original submission · Major Revisions

Based in the reviewers' comments, the manuscript must be revised.

**Language Note:** The review process has identified that the English language must be improved. PeerJ can provide language editing services - please contact us at [email protected] for pricing (be sure to provide your manuscript number and title). Alternatively, you should make your own arrangements to improve the language quality and provide details in your response letter. – PeerJ Staff

Reviewer 1 ·

Basic reporting

The manuscript entitled “Hydroponic and soil compound prediction during plant growth using multiscale feature fusion based convolutional autoencoder and GRU networks with enhanced optimization strategy” has been investigated in detail. The manuscript presents an innovative approach that employs deep learning to predict hydroponic and soil compound dynamics during plant growth, aiming to enhance agricultural decision-making and contribute to sustainable crop production. However, several critical flaws and areas of improvement need to be addressed before considering publication. There are some points that need further clarification and improvement:
1) The paper lacks clarity in describing the proposed method. Key aspects, such as the specific architecture of the Multi-Scale feature fusion-based Convolution Autoencoder with a Gated Recurrent Unit (MS-CAGRU) network, are not adequately explained.
2) Terminology inconsistencies and ambiguous phrases throughout the manuscript hinder understanding.

Experimental design

The methodology section lacks sufficient detail on data preprocessing, model architecture, hyperparameter tuning, and evaluation metrics.

The use of online data sources for model training raises concerns regarding data quality, representativeness, and potential biases.

The evaluation of the proposed model's performance is superficial and lacks rigor. Detailed experimentation, including comparisons with baseline models, is necessary to demonstrate the effectiveness of the proposed approach.

The manuscript fails to provide insights into the model's generalization capabilities, robustness, and scalability.

Validity of the findings

The rationale behind selecting specific algorithms and techniques, such as the Iteration-assisted Enhanced Mother Optimization Algorithm (IEMOA), requires clarification and justification.

“Simulation Findings and Discussions” section should be edited in a more highlighting, argumentative way. The author should analysis the reason why the tested results is achieved.

The paper lacks a comprehensive discussion on the advantages and limitations of deep learning-based approaches compared to traditional modeling methods.

It will be helpful to the readers if some discussions about insight of the main results are added as Remarks.

Additional comments

The manuscript entitled “Hydroponic and soil compound prediction during plant growth using multiscale feature fusion based convolutional autoencoder and GRU networks with enhanced optimization strategy” has been investigated in detail. The manuscript presents an innovative approach that employs deep learning to predict hydroponic and soil compound dynamics during plant growth, aiming to enhance agricultural decision-making and contribute to sustainable crop production. However, several critical flaws and areas of improvement need to be addressed before considering publication. There are some points that need further clarification and improvement:
1) The paper lacks clarity in describing the proposed method. Key aspects, such as the specific architecture of the Multi-Scale feature fusion-based Convolution Autoencoder with a Gated Recurrent Unit (MS-CAGRU) network, are not adequately explained.
2) Terminology inconsistencies and ambiguous phrases throughout the manuscript hinder understanding.
3) The methodology section lacks sufficient detail on data preprocessing, model architecture, hyperparameter tuning, and evaluation metrics.
4) The use of online data sources for model training raises concerns regarding data quality, representativeness, and potential biases.
5) The evaluation of the proposed model's performance is superficial and lacks rigor. Detailed experimentation, including comparisons with baseline models, is necessary to demonstrate the effectiveness of the proposed approach.
6) The manuscript fails to provide insights into the model's generalization capabilities, robustness, and scalability.
7) The rationale behind selecting specific algorithms and techniques, such as the Iteration-assisted Enhanced Mother Optimization Algorithm (IEMOA), requires clarification and justification.
8) “Simulation Findings and Discussions” section should be edited in a more highlighting, argumentative way. The author should analysis the reason why the tested results is achieved.
9) The paper lacks a comprehensive discussion on the advantages and limitations of deep learning-based approaches compared to traditional modeling methods.
10) It will be helpful to the readers if some discussions about insight of the main results are added as Remarks.
This study may be proposed for publication if it is addressed in the specified problems.

Reviewer 2 ·

Basic reporting

no comment

Experimental design

no comment

Validity of the findings

no comment

Additional comments

The article addresses the urgent problem of predicting the composition of hydroponics and soil to improve decision-making in crop production. The authors consider interesting methods of analysis and forecasting, which is confirmed by the relevant source materials and graphic material. The article meets the requirements of the journal and can be published in its current state.

Reviewer 3 ·

Basic reporting

This paper devises an innovative approach that leverages deep learning to predict hydroponic and soil compound dynamics during plant growth. This method not only enhances the understanding of how plants interact with their environment but also aids in making more informed decisions about agriculture, ultimately contributing to more sustainable and efficient crop production. The data needed to perform the developed hydroponic and soil compound prediction model is acquired from online resources. After that, this data is forwarded to the feature extraction phase. The weighted features, Deep Belief Network (DBN) features, and the original features are achieved in the feature extraction stage. To get the weighted features, the weights are optimally obtained using the Iteration-assisted Enhanced Mother Optimization Algorithm (IEMOA). Subsequently, these extracted features are fed into the Multi-Scale feature fusion-based Convolution Autoencoder with a Gated Recurrent Unit (MS-CAGRU) network for hydroponic and soil compound prediction. Thus, the hydroponic and soil compound prediction data is attained in the end. Finally, the performance evaluation of the suggested work is conducted and contrasted with numerous conventional models to showcase the system efficacy.
The paper needs some crucial changes and must be addressed.
Why is the title too lengthy? This makes it difficult to grasp the original contributions of this work.
The introduction is poorly written without any background information, paragraph structure, and motivation behind this work.
The methodology is well written but symbols need to be verified again. Moreover, the paragraph structure and flow need updates.
The formatting of the algorithm needs major revisions. It seems poorly written.
All equations possible need to be cited in the text.
Why have the authors considered too many equations in the manuscript? Are these aligned with the text and discussion? If yes, there needs to be explicitly discussed.
Why are results not compared with recent benchmarks instead of well-known algorithms?
The paper needs major changes in discussions.
The writing needs updates and the structure needs major changes.

Experimental design

Included in basic reporting.

Validity of the findings

Included in basic reporting.

---

## Round 0.2 · accepted · Accept

Based on the reviewer comments, the manuscript can be accepted.

Reviewer 1 ·

Basic reporting

My comments have been addressed. It is acceptable in the present form.

Experimental design

My comments have been addressed. It is acceptable in the present form.

Validity of the findings

My comments have been addressed. It is acceptable in the present form.

Reviewer 3 ·

Basic reporting

The authors have carefully revised the previous comments and improved the article accordingly.

Experimental design

N/A

Validity of the findings

N/A